# Downregulation of miR-192 Alleviates Oxidative Stress-Induced Porcine Granulosa Cell Injury by Directly Targeting Acvr2a

**DOI:** 10.3390/cells11152362

**Published:** 2022-08-01

**Authors:** Jiaqing Zhang, Qiaoling Ren, Junfeng Chen, Lingyan Lv, Jing Wang, Ming Shen, Baosong Xing, Xianwei Wang

**Affiliations:** 1Henan Key Laboratory of Farm Animal Breeding and Nutritional Regulation, Institute of Animal Husbandry and Veterinary Science, Henan Academy of Agricultural Sciences, Zhengzhou 450002, China; zhangjq@hnagri.org.cn (J.Z.); renql@163.com (Q.R.); afeng008@163.com (J.C.); wangjing@hnagri.org.cn (J.W.); 2Animal Husbandry Research Institute of Guangxi, Nanning 530001, China; llyan159@163.com; 3College of Animal Science and Technology, Nanjing Agricultural University, Nanjing 210095, China; shenm2015@njau.edu.cn; 4Henan Provincial Animal Husbandry General Station, Zhengzhou 450008, China

**Keywords:** oxidative stress, granulosa cell apoptosis, miRNA, GSPB2

## Abstract

Follicular atresia is primarily caused by breakdown to granulosa cells (GCs) due to oxidative stress (OS). MicroRNAs (miRNAs) elicit a defense response against environmental stresses, such as OS, by acting as gene-expression regulators. However, the association between miRNA expression and OS in porcine GCs (PGCs) is unclear. Here, we examined the impact of H_2_O_2_-mediated OS in PGCs through miRNA-Seq. We identified 22 (14 upregulated and 8 downregulated) and 33 (19 upregulated and 14 downregulated) differentially expressed miRNAs (DEmiRNAs) at 100 μM and 300 μM H_2_O_2_, respectively, compared with the control group. Among the DEmiRNAs, mi-192 was most induced by H_2_O_2_-mediated OS, and the downregulation of miR-192 alleviated PGC oxidative injury. The dual-luciferase reporter assay results revealed that miR-192 directly targeted Acvr2a. The Acvr2a level was found to be remarkably decreased after OS. Furthermore, grape seed procyanidin B2 (GSPB2) treatment significantly reduced the H_2_O_2_-induced upregulation of miR-192, and decreased PGC apoptosis and oxidative damage. Meanwhile, GSPB2 prevented an H_2_O_2_-induced increase in caspase-3 activity, which was enhanced by the application of the miR-192 inhibitor. These results indicate that GSPB2 protects against PGC oxidative injury via the downregulation of miR-192, the upregulation of Acvr2a expression, and the suppression of the caspase-3 apoptotic signaling pathway.

## 1. Introduction

The ovarian follicles constitute oocytes, which are structurally connected to specialized somatic cells called granulosa cells (GCs) [1]. The GCs facilitate the growth of ovarian follicles by supporting the development of oocytes, and the production of sex steroids and other growth factors. GC dysfunction has been associated with several ovarian malfunctions, including polycystic ovarian syndrome (PCOS), premature ovarian insufficiency (POI), and premature ovarian failure (POF) [2]. Under aerobic conditions, GCs experience an oxygen paradox. GCs require oxygen to maintain their function [3]. In ovarian follicles, low levels of reactive oxygen species (ROSs), such as superoxide ion (O^2−^), singlet oxygen (1O_2_), hydroxyl radical (OH∙), ozone (O_3_), hydrogen peroxide (H_2_O_2_), etc., facilitate GC signaling and homeostasis [4]. However, environmental stress and other pathological conditions can promote a dramatic enhancement in the ROS levels, resulting in severe cellular dysfunction, such as enzyme inactivation, mitochondrial abnormalities, and DNA fragmentation [5,6]. Thus, an overproduction of ROSs and/or a deficiency in antioxidants cause an imbalance in ROS-related regulatory mechanisms, resulting in oxidative stress (OS) in ovarian follicles [7]. Previous studies have shown a link between OS and ovarian toxicity in a rodent model exposed to diverse stimuli, such as gamma irradiation, chemotherapeutic drugs, or polycyclic aromatic hydrocarbons [8,9]. Additionally, ROSs derived from oxidants, such as methoxychlor, nicotine, H_2_O_2_, D-galactose, and 3-nitropropionic acid, have been found to induce mouse GC apoptosis in antral follicles [8,10].

Previous studies have shown that OS may lead to decreased reproductive efficiency of sows and delayed puberty in gilts. Reproductive disorders, including abnormal embryonic development, anovulation, etc., cause the culling of approximately half of the total number of sows culled annually. On commercial farms, there are many stressful stimuli, such as pregnancy, lactation, weaning, and high stocking density, which induce a rapid increase in ROS levels and cause OS in the maternal body. Furthermore, OS is known to promote reproductive disorders in replacement gilts and highly prolific sows [11,12]. OS plays a role in pathological progression by influencing multiple physiological processes from follicle development to oocyte maturation. Previous studies have indicated that high levels of ROS regulate the expression of epigenetic and transcriptional factors [6,10].

MicroRNAs (miRNAs) are small, noncoding RNAs (approximately 19 to 24 nucleotides) that function as gene-expression regulators and modulate cellular differentiation, proliferation, and apoptosis [13]. Recently, the aberrant expression of miRNAs has been corroborated in GC pathophysiology and ovarian endocrine abnormalities [14]. Our previous studies have shown that oxidant (H_2_O_2_, 3-NP, and diquat)-induced elevation in ROS levels promote cytotoxicity and cell death in GCs [5,9]. Additionally, recent studies have demonstrated that specific miRNAs are involved in OS-induced GC apoptosis [6,15]. However, the regulation of miRNA expression by OS in porcine GC (PGC) in unclear. In the present study, H_2_O_2_ was used to stimulate porcine GCs (PCGs) as an OS cell model, and the potential interaction of differentially expressed miRNAs (DEmiRNAs) with targeted mRNAs in H_2_O_2_-induced PGCs was analyzed using bioinformatics methods. The results will enhance our understanding of the potential biological relevance of miRNAs in PGCs exposed to OS.

## 2. Materials and Methods

### 2.1. Animals

Ovarian samples were obtained as described previously [1]. Animal procedures were performed in compliance with the guidelines of the Institutional Animal Care and Use Committee of Animal Husbandry and Veterinary Science, Henan Academy of Agricultural Sciences. The approval number for this study is IACUC-20190820002.

### 2.2. Cell Culture and Reagents

GCs, collected from healthy follicles (diameter: 3–5 mm), were cultured at 37 °C in T25 flasks containing DMEM/F-12 medium (Gibco, Grand Island, NY, USA) 10% FBS, and 1% penicillin-streptomycin (Gibco, Grand Island, NY, USA) in 5% CO_2_. For drug administration, GCs were treated with 0 to 1000 μM H_2_O_2_ for 12 h. The following miRNAs were transfected GCs: miR-192 mimics, miR-192 inhibitor, negative control (NC) mimics, and inhibitor NC (Shanghai Gene Pharma, Shanghai, China); this was achieved using Lipofectamine reagent (Invitrogen, Shanghai, China) according to the manufacturer’s protocol. 

### 2.3. GC Viability Assay

A Cell Counting Kit 8 assay (CCK-8; DOJINDO Laboratories, Japan) was used to measure GC viability following the protocol. GCs were cultured in 96-well plates until they were 90% confluent (5 × 10^4^ cells/well), followed by treatment with 0 to 500 μM H_2_O_2_ for 12 h. Post-treatment, in each well, the CCK-8 solution (10 µL) and 100 µL medium were incubated for 2 h at 37 °C. A microplate reader was used to estimate GC viability at 450 nm (Bio-Rad, Hercules, CA, USA).

### 2.4. Detection of Intracellular ROS

ROS levels in porcine GCs were measured using the GENMED intracellular-ROS red-fluorescence determination kit (GENMED, GMS10111.1; Shanghai, China). The detailed procedure was performed according to the manufacturer’s instructions. ROS levels in porcine cells were determined by measuring the oxidative conversion of dihydroethidium bromide into ethidium bromide, which emits red fluorescence when bound to DNA in the nuclei. Images were taken using an Olympus IX-73 fluorescence microscope (Olympus, Tokyo, Japan). The fluorescence intensity was evaluated in each GC using ImageJ 1.42q software (NIH).

### 2.5. Apoptosis Analysis

The TUNEL kit (Roche Applied Science, Penzberg, Germany) was used for analyzing GC apoptosis following the manufacturer’s guidelines. The Olympus IX-73 fluorescence microscope was used to capture images. A total of five fields of vision were selected to count the numbers of apoptosis cells and total cells. Then, the rate of apoptosis of PGCs was calculated. All experiments were repeated at least 3 times.

### 2.6. Caspase-3 Activity Assay

Caspase-3 activity was examined using caspase-3 activity assay kits (Beyotime Institute of Biotechnology) according to the manufacturer’s instructions. Briefly, the GCs were homogenized in 10 mL reaction buffer containing 1 mL Caspase-3 fluorogenic substrate Ac-DEVD-pNA (acetylAsp-Glu-Val-Asp p-nitroanilide) and incubated for 2 h at 37 °C. Cleavage of the substrates was measured at 405 nm using a microplate reader.

### 2.7. Small-RNA Library Construction and Sequencing

The TRIzol reagent was used to isolate total RNA following the specified protocol. The Bioanalyzer 2100 (Agilent, Santa Clare, CA, USA) with a RIN > 7.0 was used for analyzing the purity and quantity of the extracted RNA. A small-RNA library was built using the TruSeq Small-RNA Sample Prep Kit (Illumina, San Diego, CA, USA) using total RNA (1 µg). Then, the Illumina HiSeq 2500 was used to perform single-end sequencing (36 bp or 50 bp) at LC-BIO (Hangzhou, China).

### 2.8. Data Analysis

The raw reads were subjected to ACGT101-miR (LC Sciences, Houston, TX, USA), an in-house program, to remove junk, adapter dimers, common RNA families (rRNA, tRNA, snRNA, and snoRNA), low complexity, and repeats. Next, we used BLAST to map unique 18- to 26-nucleotide-long sequences to specific precursors in miRBase 21.0 to detect the known miRNAs (in the hairpin arms) and novel 3p-derived and 5p-derived miRNAs (in the arm opposite the annotated mature miRNA-containing arm). The alignment parameters allowed length variations at both the 5′ and 3′ ends, as well as one sequence mismatch. The known miRNAs were classified as the unique sequences that mapped mature miRNAs in hairpin arms to specific species. Next, we used BLAST to map the remaining sequences to other species precursors, except for certain species, in the miRBase 21.0; this was followed by estimation of their genomic location by mapping the pre-miRNAs against the specific species genomes using BLAST. These two categories were classified as the known miRNAs. Next, using BLAST software, we aligned the unmapped sequences against the specific genomes and used the RNAfold software to predict the hairpin RNA structures containing sequences from the flank 80 nt sequences. The secondary structure was predicted based on the following criteria: (1) less than or equal to 12 nucleotides in one bulge in the stem; (2) cutoff of free energy (kCal/mol ≤ −15); (3) length of the hairpin loop (≤20); (4) less than or equal to 4 biased errors in one bulge in the mature region; (5) less than or equal to 7 errors in mature region; (6) greater than or equal to 80% maturity in the stem; (7) more than 16 base pairs in the stem region of the predicted hairpin; (8) length of the hairpin (up and down stems + terminal loop ≥ 50); (9) less than or equal to 8 nucleotides in one bulge in the mature region; (10) less than or equal to 2 biased bulges in the mature region; (11) more than or equal to 12 base pairs in the mature region of the predicted hairpin. All data have been deposited into the NCBI Gene Expression Omnibus and are accessible via GEO series accession number GSE201369.

### 2.9. Analysis of Differentially Expressed miRNAs

A χ^2^-squared 2 × 2 test, a Student’s *t*-test, an ANOVA, a Fisher’s exact test, or an nXn test were used to examine the differential expression of miRNAs based on normalized deep-sequencing counts. For each test, the limit of statistical significance was set between 0.01 and 0.05.

### 2.10. The Prediction of Target Genes of miRNAs

We used miRanda 3.3a and TargetScan 50 to identify the miRNA binding sites to estimate the target genes of DEmiRNAs. We combined the prediction data from both algorithms to obtain the overlapping results. We also annotated the GO terms and KEGG pathways of the targets of these DEmiRNA.

### 2.11. Quantitative Real-Time RT-PCR (qRT-PCR)

Total RNA was extracted from PGCs using the TRIzol reagent according to the manufacturer’s protocol. Firstly, total RNA was reverse-transcribed using the M-MLV reverse transcriptase (Takara, Dalian, China). Then, qRT-PCR reactions were performed using a standard SYBR Green PCR kit. The 20 μL reaction mixture contained 10 μL of 2× SYBR Premix Ex Taq, 1 μL 10 μM of each primer, 2 μL of cDNA, and 7 μL of ddH_2_O. qRT-PCR for each gene was performed in triplicate for each sample using a Roche Light Cycler 480 system with SYBR Green Real-Time PCR Mix (Takara, Dalian, China). The mRNA primers for Bim, Bax, and Acvr2a were based on those used in previously published studies [16,17,18]. GAPDH was used as an internal control. The miRNA was extracted from the control and the H_2_O_2_ treatment samples (100 and 300 μM) using a miRNA extraction kit (Tiangen Biotech, Beijing, China). First, the miRcute Plus miRNA First-Strand cDNA Synthesis Kit was used to synthesize cDNA. The reaction mix (20 μL), consisting of total RNA (3 μL), 2× miRNA RT Reaction Buffer (10 μL), miRNA RT Enzyme Mix (2 μL), and ddH2O (5 μL), was incubated for 1 h at 42 °C, followed by 3 min at 95 °C for enzyme inactivation. The miRcute Plus miRNA qPCR Detection Kit (SYBR Green) (Tiangen Biotech, Beijing, China) was used to quantify miRNA. Primers for miR-181d-5p (cat. no. CD201-T), miR-23b-5p (cat. no. CD201-0311), miR-129-1-3p (cat. no. CD201-0200), miR-19b-3p (cat. no. CD201-0278), miR-190b (cat. no. CD201-T), miR-375-3p (cat. no. CD201-0173), miR-192 (cat. no. CD201-0082), miR-339-5p (cat. no. CD201-T) and U6 (cat. no. CD201-0145) were acquired from Tiangen Biotech Co., Ltd (Beijing, China). The relative miRNA expression levels were derived using the 2^−ΔΔCt^ method [19].

### 2.12. Western Blot Analysis

Total proteins were prepared using protein lysis buffer and quantified using a BCA protein assay kit (Beyotime, Beijing, China). Next, 20 μg of total protein/sample was loaded onto a 12% SDS-polyacrylamide gel and transferred onto a PVDF membrane (Millipore, Billerica, MA, USA) via electroblotting. Non-specific binding sites were blocked with 5% bovine serum albumin in TBST for 1.5 h. Then, the membranes were incubated with primary antibodies against ACVR2A (DF6733, Affinity Biosciences, Cincinnati, OH, USA), Cleaved-Caspase 3 (AF7022, Affinity Biosciences, Cincinnati, OH, USA), and GAPDH (BM1623, Boster Biological Technology, Wuhan, China) overnight at 4 °C; all primary antibodies were used at a dilution of 1:200 in PBS containing 5% bovine serum albumin. Immunoreactivity was detected via treatment with an appropriate HRP-conjugated secondary antibody. Protein bands were visualized by exposing the blots to an enhanced chemiluminescence detection system (LAS-4000 imager, Fujiflm, Tokyo, Japan).

### 2.13. Statistical Analysis

SPSS v17.0 (SPSS, Armonk, NY, USA) was used for data analyses, and the results are represented as the means ± standard error (S.E.). *p*-values of less than 0.05 and 0.01 were considered significant and extremely significant differences, respectively.

## 3. Results

### 3.1. OS Reduces Viability and Induces Apoptosis in PGCs

The primary cultured PGCs were treated with various concentrations of H_2_O_2_ (0, 50, 100, 200, 300, 500, and 1000 μM) for 12 h to establish an in vitro OS model. Next, we conducted a CCK-8 assay to examine GC proliferation and viability. H_2_O_2_ induced a significant time-dependent reduction in cell viability (Figure 1A). At 100 μM H_2_O_2_, the experimental group demonstrated considerably lower cell viability than the control group (*p* < 0.05). Furthermore, at higher concentrations of H_2_O_2_, a substantial reduction in cell viability was observed (*p* < 0.01). Thus, doses of 100 μM and 300 μM H_2_O_2_ were chosen as the preferred doses for subsequent experiments.

Next, cultured GCs were treated with 0, 100, and 300 μM H_2_O_2_ for 12 h, followed by a TUNEL assay to examine the rate of apoptosis. The rate of apoptosis increased with an increase in H_2_O_2_ concentration (Figure 1B,C). At 100 μM H_2_O_2_, a considerable increase in GC apoptosis was observed (*p* < 0.05), which further increased at 300 μM H_2_O_2_. These data suggest that H_2_O_2_ induced dose-dependent apoptosis of PGC, which resulted in reduced cell viability.

### 3.2. Identification of H_2_O_2_-Induced Changes in miRNA Expression in PGCs

We treated PGCs with H_2_O_2_ to develop a cellular model of OS to understand the impact of H_2_O_2_-induced OS on miRNAs. Based on the above results, 100 μM and 300 μM H_2_O_2_ were chosen as the low concentration (LOS) and the high concentration (MOS) for miRNA profiling, respectively. After treating PGCs for 12 h, the miRNA profiles were analyzed using a high-throughput sequencing strategy. Overall, 22 DEmiRNAs (14 upregulated and 8 downregulated) were identified between the LOS group at |log2 (foldchange)| ≥1 and *p*-value < 0.05 and the control group (CON) (Figure 2A). Furthermore, 33 DEmiRNAs (19 upregulated and 14 downregulated) were identified between the MOS group and the CON group under identical criteria (Figure 2B). Appendix A lists all the DEmiRNAs. These data indicate that a low concentration of H_2_O_2_ had a minor effect on the miRNA expression of PGCs, while a high concentration of H_2_O_2_ could affect the expression of several miRNAs. To validate the accuracy of the sequencing data, eight miRNAs with significantly altered expression were selected for RT-PCR detection. The results agree with the sequencing data, suggesting the high accuracy of our sequencing analysis (Figure 2C).

### 3.3. Functional Analysis of DEmRNAs

The miRNAs play their functional roles by binding to the complementary site in the 3′-untranslated regions of their target mRNAs. The target genes of H_2_O_2_-sensitive miRNAs were analyzed using miRanda and TargetScan. The results revealed that LOS-induced miRNAs targeted 14,244 genes (Appendix A), and MOS-induced miRNAs targeted 20,568 genes (Appendix A). Next, we performed KEGG and GO enrichment analyses to detect the cellular functions of these genes in PGCs under OS. The top 20 gene clusters involving biological processes and pathway analyses sensitive to low concentrations of H_2_O_2_ are summarized in Figure 3A,B. The most significant biological processes included transcriptional regulation, signal transduction, DNA-templated, positive regulation of transcription from the RNA polymerase II promoter, transport, and the oxidation-reduction process. The most significant pathways were the pathways in cancer, HTLV-I infection, Ras, and Rap1, and the endocytosis signaling pathways. The top 20 gene clusters involving biological processes and pathway analyses sensitive to high concentrations of H_2_O_2_ are summarized in Figure 3C,D. The most significant biological processes were transcriptional regulation, DNA-templating, the positive regulation of transcription from the RNA polymerase II promoter, signal transduction and transport, and the oxidation-reduction process. The most significant pathways were the pathways in cancer, MAPK, Ras, HTLV-I infection, and Rap1 signaling.

Then, we selected overlapping miRNAs between the two concentration groups and further constructed the miRNA–KEGG networks, miRNA–gene networks, and miRNA–GO networks to detect the primary regulatory functions of the overlapping miRNAs and their target genes. We identified 34 pathways by the predicted target genes of 13 DEmiRNAs (Figure 4).

Each signaling pathway was co-regulated by more than three miRNAs. The specific miRNAs that co-regulated these pathways are shown in Appendix A. As shown in Figure 5 and Appendix A, hsa-miR-141-3p_R+1, PC-5p-9551_196, and hsa-miR-222-5p_L+2R-1, with the target gene numbers 44, 39, and 32, respectively, were the top three miRNAs. Thirteen miRNAs (hsa-miR-141-3p_R+1, PC-5p-9551_196, hsa-miR-222-5p_L+2R-1, hsa-miR-4792_1ss9GT, sha-mir-24-1-p3_1ss2GC, ssc-miR-139-3p, ssc-miR-1839-3p_R+2, ssc-miR-190b, ssc-miR-192, ssc-miR-194a_R+2, ssc-miR-194b-5p_1ss10GA, ssc-miR-215_R+1, and ssc-mir-4332-p5_1ss18CA) were predicted to be involved in apoptosis and autophagy. The top 10 target genes were NR5A1 (nuclear receptor subfamily 5 group A member 1), TGF-ß-R1 (transforming growth factor-beta receptor 1), VEGFC (vascular endothelial growth factor C), WNT5A (Wnt family member 5A), RAC1 (Rac family small GTPase 1), MRAS (muscle RAS oncogene homolog), ITGB1 (integrin subunit beta 1), ACVR1B (activin receptor IB), WNT2B (Wnt family member 2B), and SMAD3 (SMAD family member 3).

The DEmiRNAs played vital roles in several biological processes, including the activation of MAPK activity (GO:0000187), the apoptotic process (GO:0006915), the cellular response to DNA damage stimulus (GO:0006974), the transforming growth factor-beta receptor signaling pathway (GO:0007179), and the positive regulation of cell population proliferation (GO:0008284) (Figure 6 and Appendix A). Thus, these results provide us with more comprehensive information on the biological effect of H_2_O_2_ treatment in PGCs.

### 3.4. miR-192 Overexpression Promotes Porcine GCs Apoptosis

Previous studies showed that miR-192 expression is significantly upregulated in human endothelial cells [20] and animal ovarian tissues [21] under stress. GC apoptosis is the main cause of follicular atresia. Therefore, the role of miR-192 in this process was further studied. The cultured GCs were transfected with miR-192 mimics or mimics NC. Analysis of cell viability showed that the GC proliferation rate was decreased significantly after transfection with the miR-192 mimics (*p* < 0.05; Figure 7A). Furthermore, the number of GC apoptosis was increased significantly after transfection with the miR-192 mimics when compared to those transfected with the mimics NC (*p* < 0.05; Figure 7B,C). It was recently reported that Bim, a BH3-only member of the BCL-2 family, can induce apoptosis by inactivating anti-apoptotic BCL-2 family members and by promoting Bax activation, which subsequently activates caspase-3-mediated apoptosis [22]. As expected, the mRNA expression levels of the apoptotic-related genes (Bax and Bim) were significantly higher in the GCs transfected with the miR-192 mimics than in those transfected with the mimic NCs (*p* < 0.05; Figure 7D,E). The activity assay suggested that the transfection of miR-192 mimics caused a significant increase in GCs compared with mimics NC (*p* < 0.05; Figure 7F). These results indicate that miR-192 overexpression strongly promoted PGC apoptosis.

### 3.5. miR-192 Knockdown Counteracts H_2_O_2_-Induced PGC Injury

Recent studies have demonstrated that miR-192 is involved in the regulation of apoptotic signaling [20]. To further validate whether H_2_O_2_-induced damage affects the expression of endogenous miR-192 in GCs, the GCs were treated with various concentrations of H_2_O_2_ (0, 50, 100, 250, and 500 μM) for 12 h. The expression level of miR-192 was increased significantly by H_2_O_2_ in a dose-dependent manner (Figure 8A). To assess the effects of miR-192 knockdown on miR-192 expression, cell viability, intracellular ROS levels, and caspase-3 activity, the cultured primary cells was investigated in vitro. The expression levels of miR-192 were significantly decreased in the miR-192 inhibitor group compared with the inhibitor control group, and H_2_O_2_ exposure significantly increased miR-192 expression in the inhibitor control group compared with the expression levels in the miR-192 inhibitor group (*p* < 0.05; Figure 8B). Furthermore, miR-192 knockdown partially rescued H_2_O_2_-induced GC viability loss, when compared with the H_2_O_2_-treated group (*p* < 0.05; Figure 8C). In addition, the cultured GCs transfected with miR-192 inhibitor and treated with H_2_O_2_ appeared to be less susceptible to H_2_O_2_-induced injury, as indicated by the lower caspase-3 activity and intracellular ROS levels compared with the H_2_O_2_-treated inhibitor control group (*p* < 0.05; Figure 8D–F). These results suggest that miR-192 knockdown in porcine GCs may exert a protective effect against H_2_O_2_-induced injury.

### 3.6. Acvr2a Is a miR-192 Target

Among DEmiRNAs, we found that miR-192 was the most significantly upregulated miRNA in PGCs after qRT-CR validation. Pathway analysis showed that the miR-192 targets are enriched in steroid biosynthesis, the insulin signaling pathway, and the GnRH signaling pathways (Figure 9A). Gene ontology analysis indicated that the miR-192 targets were mainly involved in the regulation of the digestive-system process, the rRNA catabolic process, and RNA surveillance (Figure 9B). Acvr2a was identified as a putative miR-192 target; this gene is widely expressed in ovarian GCs, and it is closely related to GC proliferation and follicular development. Therefore, Acvr2a was chosen as a candidate target gene for further study. Bioinformatics analysis identified a putative miR-192 binding site in the 3′UTR of Acvr2a (Figure 9C). To confirm that Acvr2a is a direct target of miR-192, the WT and mutant vectors were co-transfected HEK293T cells with either miR-192 mimics or mimics NC, and their luciferase activity was measured. The results showed that the miR-192 mimic, but not the mimics NC, markedly suppressed the luciferase activity of the Acvr2a 3′UTR-WT but not that of Acvr2a 3′UTR-MT, suggesting that miR-192 directly targets the 3′-UTR of Acvr2a (Figure 9D). To validate that the Acvr2a is a target of miR-192, the mRNA and protein expression levels of Acvr2a were determined after the transfection of the GCs with the miR-192 mimics or mimics NC. The results indicate that the expression levels of Acvr2a mRNA and the ACVR2A protein markedly decreased in GCs transfected with the miR-192 mimics (*p* < 0.05; Figure 9E,F). These data suggest that Acvr2a is a validated target of miR-192 in PGCs.

### 3.7. miR 192a Knockdown Inhibits PGC Oxidative Damage through the Upregulation of Acvr2a and Suppression of Caspase-3 Activity

ACVR2A expression regulation was previously found to be closely associated with GC proliferation [23]. Therefore, we further studied whether miR-192 knockdown may exert a protective effect against H_2_O_2_-induced PGC apoptosis. As shown in Figure 10A,B, the expression levels of Acvr2a mRNA and the ACVR2A protein were significantly downregulated in the inhibitor control plus H_2_O_2_-treated PGCs compared with H_2_O_2_-free PGCs, and this decreased expression could be significantly increased by miR-192 knockdown. In contrast, PGCs transfected with the miR-192 inhibitor increased their expression levels of Acvr2a mRNA and the ACVR2A protein in the H_2_O_2_-induced PGCs. Furthermore, the caspase-3 mRNA expression and cleaved caspase-3 protein level were increased in the inhibitor control plus H_2_O_2_-treated PGCs compared with H_2_O_2_-free PGCs; this increased expression was reduced by miR-192 knockdown (*p* < 0.05; Figure 10C,D). These results suggest that miR-192 promotes H_2_O_2_-induced PGC apoptosis by modulating the expression of Acvr2a and caspase-3.

### 3.8. Acvr2a Is Required for GSPB2-Mediated PGC Protection against Oxidative Damage

Grape seed procyanidin B2 (GSPB2), a polyphenolic component found in red wine and grapes, has been reported to protect against ovarian oxidative damage [5]. To determine whether GSPB2 regulates cell viability in cultured PGCs under oxidative stress, a CCK-8 assay was employed to determine PGC viability. As shown in Figure 11A,B, H_2_O_2_ significantly reduced PGC viability in a dose-dependent manner, and PGCs pretreated with GSPB2 displayed a marked increase in GSPB2-plus-H_2_O_2_-treated PGCs compared with H_2_O_2_-treated PGCs. The inhibition of Acvr2a expression has been suggested to inhibit GC proliferation and induce apoptosis [23]. To explore whether Acvr2a correlates with GSPB2-mediated apoptosis inhibition during oxidative stress, the transcript encoding Acvr2a was inhibited using RNA interference. As shown in Figure 11C, the expression level of Acvr2a mRNA was markedly lower in PGCs transfected with siRNA plus H_2_O_2_ than those transfected with the SC siRNA. PGCs pretreated with GSPB2 significantly increased the expression level of Acvr2a mRNA. Furthermore, Acvr2a knockdown significantly inhibited the viability of PGCs compared with those transfected with the SC siRNA. PGCs pretreated with GSPB2 exhibited an increase in viability compared with untreated control cells Figure 11D. In addition, by measuring caspase-3 activity under the same conditions (Figure 11E), we further confirmed that Acvr2a is required for GSPB2-mediated PGC protection against oxidative injury.

### 3.9. GSPB2 Protects PGCs from H_2_O_2_-Induced Oxidative Damage through the Downregulation of miR-192

Previous studies have revealed that GSPB2 exhibits inhibitory effects on GC oxidative damage [24]. To determine whether GSPB2 supplement affects the expression of endogenous miR-192 in PGCs, the PGCs were pretreated with GSPB2 for 12 h and, finally, incubated with H_2_O_2_ for 12 h, and the expression of miR-192 was determined via qRT-PCR. As shown in Figure 12A, PGCs treated with H_2_O_2_ significantly enhanced the expression of miR-192 compared with control group, and GSPB2 supplement significantly suppressed the H_2_O_2_-induced upregulation of miR-192 expression in PGCs (*p* < 0.05). Cell viability assays showed that the transfection of the miR-192 inhibitor resulted in the upregulation of PGC viability induced by H_2_O_2_, and the transfection of the miR-192 mimics resulted in the downregulation of PGC viability induced by H_2_O_2_ (*p* < 0.05) (Figure 12B,D). Furthermore, the caspase-3 activity assays indicated that transfection of the miR-192 inhibitor caused the downregulation of caspase-3 activity in H_2_O_2_-treated PGCs (*p* < 0.05). In contrast, PGCs transfected with miR-192 mimics resulted in the upregulation of caspase-3 activity in H_2_O_2_-treated PGCs (*p* < 0.05) (Figure 12C,E). These results indicate that GSPB2 suppressed H_2_O_2_-induced PGC oxidative injury by regulating miR-192 expression.

## 4. Discussion

The ovarian follicle is a primary site for the production of steroid hormones and is vital for normal ovarian function and female reproduction. Although the mammalian ovaries contain numerous preantral follicles at different stages, more than 99% of the follicles are destroyed through follicular atresia, which has been found to be the primary cause of GC apoptosis. OS, which is caused by the overproduction of ROSs and an impaired antioxidant defense system, is also involved in GC apoptosis. Several factors, such as injury, inflammation, smoking, nutrient loss, and aging, are known to accelerate GC apoptosis, resulting in follicular atresia [8]. On the contrary, OS inhibitors suppress GC apoptosis during follicular atresia. For example, OS-induced apoptosis in preovulatory follicles was blocked by treatment with FSH and was related to enhanced GSH synthesis [25]. Proanthocyanidins may protect humans and PGCs against OS-induced apoptosis [24,26]. As master regulators of gene expression, miRNAs may play a significant role in OS-induced GC death. Here, we found that H_2_O_2_ (100 μM and 300 μM) could induce ROS generation, promote cell apoptosis, and decrease cell viability in vitro.

miRNAs function as post-transcriptional gene-expression regulators by inhibiting protein translation or by targeting mRNAs for degradation. Accumulating evidence has shown that OS significantly alters miRNA expression in several cell types, including human, mouse, and bovine cells [27,28,29]. Furthermore, miRNAs have been shown to be the major regulators under conditions of OS [30]. Importantly, studies have substantiated the role of miRNAs in almost all adverse events occurring in the female reproductive system, including PCOS, POF, and POI. It has also been shown that a number of miRNAs may be involved in regulating GC proliferation (i.e., miR-145 and miR-224), differentiation (i.e., miR-224, miR-383, and miR-378), and apoptosis (i.e., miR-23a, miR-21, and miR-26b) [6,31]. In this study, we identified 22 DEmiRNAs (14 upregulated and 8 downregulated) in the LOS group and 33 DEmiRNAs (19 upregulated and 14 downregulated) in the MON group through miRNA-Seq (|log2 (fold change)| ≥ 1 and *p*-value < 0.05). Moreover, several of these miRNAs have been proven to be regulated by various stimuli in different types of cells. For example, a higher expression of miR-200c triggered by OS in vitro significantly attenuates cell growth and induces apoptosis and senescence in human umbilical vein endothelial cells [32]. miR-27a-5p has been proven to be induced by inflammatory stimuli, such as Ox-PAPC, IL-1β and TNFα, which subsequently modulate the extent of NF-kB signaling in HUVECs [33]. The miR-375 levels were significantly upregulated in doxorubicin-treated rat and mouse cardiomyocytes, and the inhibition of miR-375 reduced inflammation and increased the survival of cardiomyocytes [34]. Additionally, several DEmiRNAs, such as miR-139-3p, miR-190b, and miR-19b, are known to be involved in proliferation, growth, aging and apoptosis in HeLa cells, follicular fluid, and SH-SY5Y cells by targeting NOB1, EXT1, and HAPLN4, respectively [35,36,37].

Additionally, OS may trigger or mediate multiple signaling pathways to regulate cellular processes, such as the PI3K-Akt [38], FoxO [10], mTOR [39], NF-kB [40], and JNK signaling pathways [41]. Here, we identified the top 20 pathways in PGCs following exposure to OS through the enrichment analysis of multiple miRNA target genes (Figure 3B,D). Interestingly, we observed that the signaling pathways mentioned above also existed in our enrichment analysis. Furthermore, Ras signaling, Rap1 signaling, and pathways in cancer were also enriched. Several of these pathways play a role in cell survival, autophagy, and apoptosis. For example, the FoxO and NF-kB signaling pathways are directly involved in the survival of porcine and mouse GCs [38,39,42,43]. Moreover, PI3K-Akt activation and JNK inhibition have been shown to inhibit apoptosis in porcine and mouse GCs in response to OS [38,42,44]. Pathway analysis of our data suggests that OS might induce the initiation and development of PGC death by inhibiting the activation of pathways related to cell survival and activating proapoptotic pathways.

Previous studies have demonstrated that OS significantly alters gene expression in different types of cells [10,26,32]. Here, a network, consisting of target mRNAs and dysregulated miRNAs obtained from the LOS group and MON group versus CON, was constructed (Figure 5). Moreover, multiple hub genes, including FoxO1, TGFBR1, ACVR1B, SMAD3, SMAD7, BMPR2, XIAP, and ESR1, were identified. Among them, FoxO1, a key modulator of stress, has been proven to be regulated by H_2_O_2_-induced OS at different levels [6,10]. SMAD3 BMPR2, XIAP, and ESR1 have also been proven to be regulated by OS in skin fibroblasts [45], vascular smooth-muscle cells [46], PC6.3 cells [47], and breast cancer [48]. In addition, these genes have been reported to be major regulators affecting cell state and function. For instance, TGFB1, BMP7, BMP6, BMPR2, SMAD3, SMAD1, and SMAD7 are involved in cell proliferation and apoptosis. ACVR1B, ESR1, TGFBR1, and NR5A1 are associated with hormone secretion and cytokine responses. FoxO1, XIAP, BAD, ATG12, FAS, and GADD45B are apoptosis and autophagy factors. The functions of these miRNA target genes could partially explain the effects of H_2_O_2_-induced OS on PGC function and state.

miRNAs are known to play essential roles in several cellular processes [49,50]. In this study, we performed miRNA–GO network analysis based on the relationship between significant biological functions and miRNAs. The results showed that MAPK activity, the apoptotic process, the cellular response to DNA damage stimulus, the TGF-β receptor signaling pathway, and positive regulation of the proliferation of the cellular population were enriched in the miRNA–GO regulatory network. Moreover, these biological processes were associated with different cell types. For example, it was found that the miR-19b-induced anti-apoptosis effect was promoted by the MAPK pathway in SH-SY5Y cells [37]. miR139-3p may act as a tumor suppressor that inhibits cell migration, proliferation, and invasion, and induces cell apoptosis through the downregulation of NOB1 expression [35]. In addition, miR-192 can target the RB1 gene to induce lung cancer cell apoptosis through the caspase pathway [51]. Thus, these data indicate that H_2_O_2_-induced OS may regulate the biological functions of PGCs by altering the expression of specific miRNAs.

Recent studies suggest that miR-192 is regulated by hypoxia and oxidative stress. For instance, in the ovary of the marine medaka, miR-192 was found to be statistically significantly upregulated under hypoxia [21], suggesting that miR-192 plays a regulatory role in ovaries under hypoxic stress. In endothelial cells (ECs), Paola et al. reported that miR-192 was most induced by oxidative stress, and miR-192 overexpression significantly decreased ECs proliferation and induced EC death, indicating that miR-192 has a potential role in cell survival [20]. The results of this study show that miR-192 levels are significantly increased in the PGCs subjected to H_2_O_2_ exposure. MiR-192 knockdown enhances cell viability and reduces caspase-3 activity following H_2_O_2_ incubation (Figure 8). ACVR2A is a critical regulator of follicle development in mammals [52]. Acvr2a knockdown can lead to the inhibition of GC proliferation [23]. Our current study indicates that miR-192 overexpression reduces Acvr2a mRNA and ACVR2A protein levels in PGCs by targeting Acvr2a 3′UTR (Figure 9). Furthermore, the mRNA and protein expression levels of Acvr2a decrease in PGCs after H_2_O_2_ exposure, and miR-192 knockdown significantly enhances Acvr2a levels. These findings indicate that increases in miR-192 expression and decreases in Acvr2a expression resulting from H_2_O_2_ treatment cause PGC oxidative injury.

Our findings show that miR-192 expression is significantly upregulated in PGCs following H_2_O_2_ treatment. This finding is in line with those of previous reports [20,21]. Furthermore, miR-192 expression is markedly increased in GCs of women’s with diminished ovarian reserves [53], suggesting that it might be connected to GC proliferation and survival. Therefore, we further studied the role of miR-192 in H_2_O_2_-triggered PGC damage. Using overexpression, knockdown, bioinformatic analysis, luciferase assays, and mRNA and protein expression analyses, we confirmed that Acvr2a is a target of miR-192. Acvr2a is the critical receptor type II A of activin, a member of the transforming growth factor-beta (TGF-β) superfamily through which activin or related TGF-β ligands induce FSH production [54]. In mice, the inhibition of Acvr2a expression results in GC apoptosis [23]. Furthermore, targeted depletion of Acvr2a in mice increased the number of atretic follicles [55]. The results herein show that Acvr2a expression is markedly decreased in H_2_O_2_-treated GC apoptosis. Furthermore, miR-192 inhibition markedly reverses the expression of Acvr2a mRNA and the ACVR2A protein during H_2_O_2_ exposure (Figure 10). These results indicate that Acvr2a is a direct target of miR-192 and that it mediates the role of miR-192 in PGC oxidative damage during H_2_O_2_ exposure. Meanwhile, all of these findings lead us to speculate that miR-192 mediates H_2_O_2_-induced PGC apoptosis by suppressing Acvr2a expression, resulting in the dephosphorylation of type I receptors and the inactivation of intracellular R-Smad signal transducers, thereby increasing the relative abundance of Bim and Bax and causing caspase-3-mediated apoptosis.

GSPB2 is one of the main components of grape seed proanthocyanidin extract (GSPE). Growing evidence suggest that GSPB2 alleviates oxidative-stress-activated GC apoptosis [24,26]. In this study, we found that PGCs pretreated with GSPB2 for 12 h dramatically inhibit the cell-viability loss triggered by H_2_O_2_. Furthermore, GSPB2 markedly inhibited caspase-3 activity induced by H_2_O_2_ in PGCs, which was associated with the increasing expression of Acvr2a (Figure 11). Here, we aimed to elucidate the key role of miR-192 in PGC injury triggered by H_2_O_2_, and whether GSPB2-mediated PGC protection is correlated with miR-192 regulation. Our data suggest that the expression level of miR-192 is markedly increased in PCs treated with H_2_O_2_, but this level is significantly decreased in PGCs pretreated with GSPB2 prior to H_2_O_2_. Moreover, miR-192 knockdown enhanced the cytoprotective role of GSPB2 against H_2_O_2_-triggered PGC damage. Collectively, these data demonstrate that the cytoprotective role of GSPB2 against H_2_O_2_-triggered PGC damage occurs partially through the downregulation of miR-192.

## 5. Conclusions

In conclusion, the depression of miR-192 alleviates oxidative stress by directly targeting Acvr2a in H_2_O_2_-caused PGC damage. miR-192/Acvr2a might act as a potential and novel therapeutic target for oxidative-stress-induced PGC injury. These results also provide important insights into the potential discovery and development of GSPB2 as a novel drug for reproductive-disease treatment.

## Figures and Tables

**Figure 1 cells-11-02362-f001:**
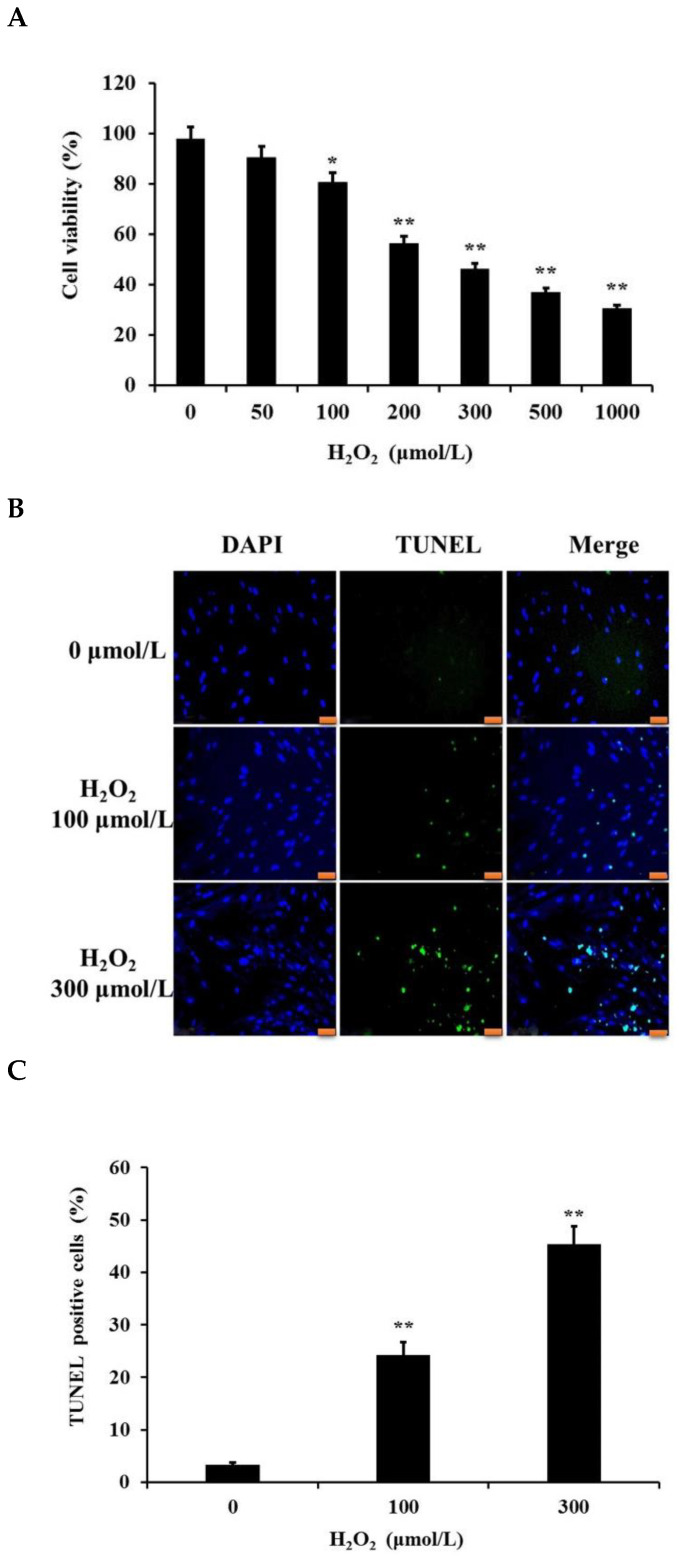
The H_2_O_2_-mediated OS model in PGCs. (**A**) Primary cultured GCs were treated with different concentrations of H_2_O_2_ for 12 h. The CCK-8 assay was used to determine GC viability. (**B**) GC apoptosis was determined via TUNEL analysis; scale bars correspond to 50 μm. (**C**) The average number of TUNEL-positive nuclei/visual fields was used to quantify GC apoptosis. Data represent mean ± S.E.; each group: *n* = 3. * *p* < 0.05, ** *p* < 0.01 compared to control group.

**Figure 2 cells-11-02362-f002:**
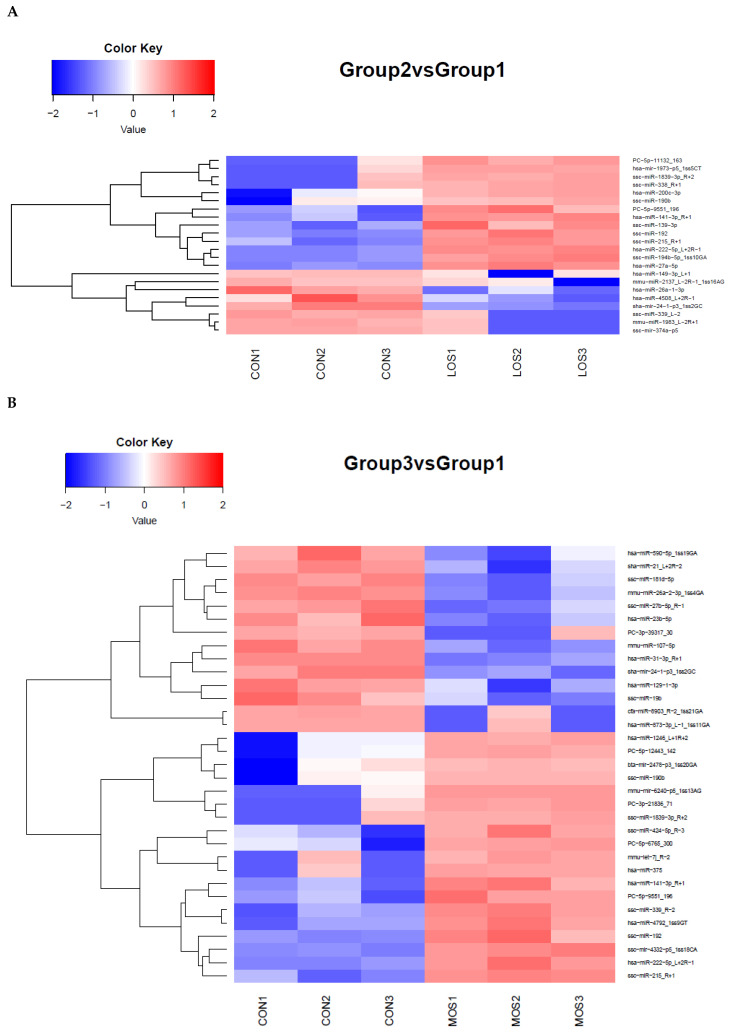
Alterations in the miRNA profiles in H_2_O_2_ treated PGCs. (**A**) The heat map of 22 DEmiRNAs between low-concentration group (LOS) and normal control (CON). The color scale of the heat map ranges from blue (low expression) to red (high expression). (**B**) The heat map of 33 DEmiRNAs between high-concentration group (MOS) and normal control (CON). Low-concentration group (LOS): 100 μM H_2_O_2_; high-concentration group (MOS): 300 μM H_2_O_2_; (**C**) DE miRNAs with high fold changes were chosen for qRT-PCR validation. Dates were normalized using the U6 gene. The data are presented as mean ± S.E. (*n* = 3). The values are shown as log2 (fold change).

**Figure 3 cells-11-02362-f003:**
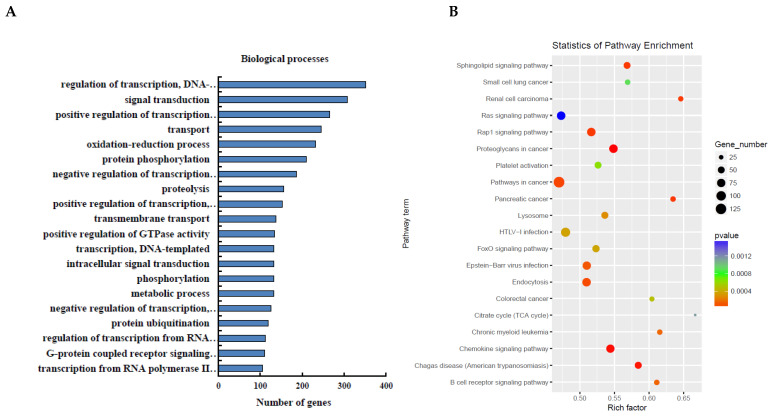
Enrichment analysis of predicted target genes of deregulated miRNAs. (**A**) Significantly altered cellular processes of predicted target genes of DEmiRNAs after H_2_O_2_ treatment (100 μM). (**B**) Significantly altered pathways of the target genes. (**C**) Significantly altered cellular processes of predicted target genes of DEmiRNAs after H_2_O_2_ treatment (300 μM). (**D**) Significantly altered pathways of the target genes.

**Figure 4 cells-11-02362-f004:**
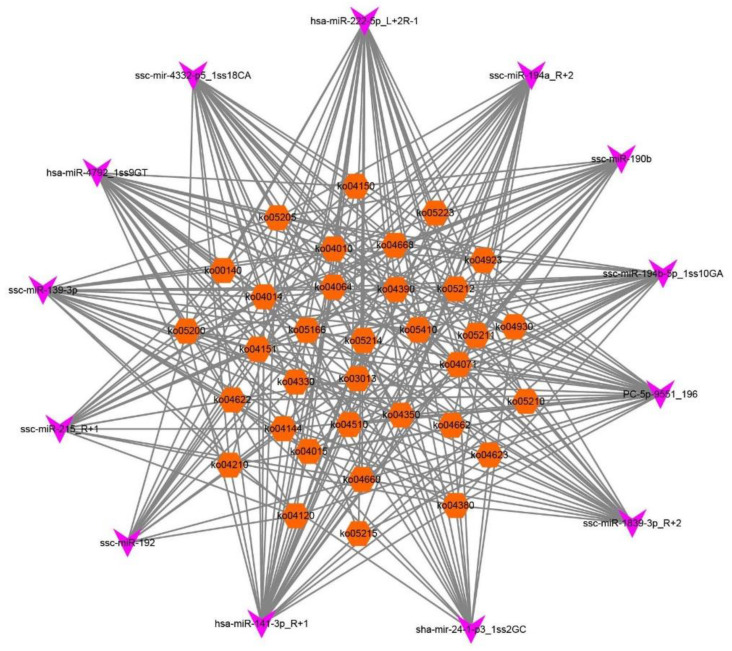
DEmiRNA–KEGG network analysis. The miRNA–KEGG network was built based on the interactions between signal pathways and their respective DEmiRNAs. For the network, the diamond nodes represent miRNA, and the orange nodes represent pathways affected by two/three DE miRNAs.

**Figure 5 cells-11-02362-f005:**
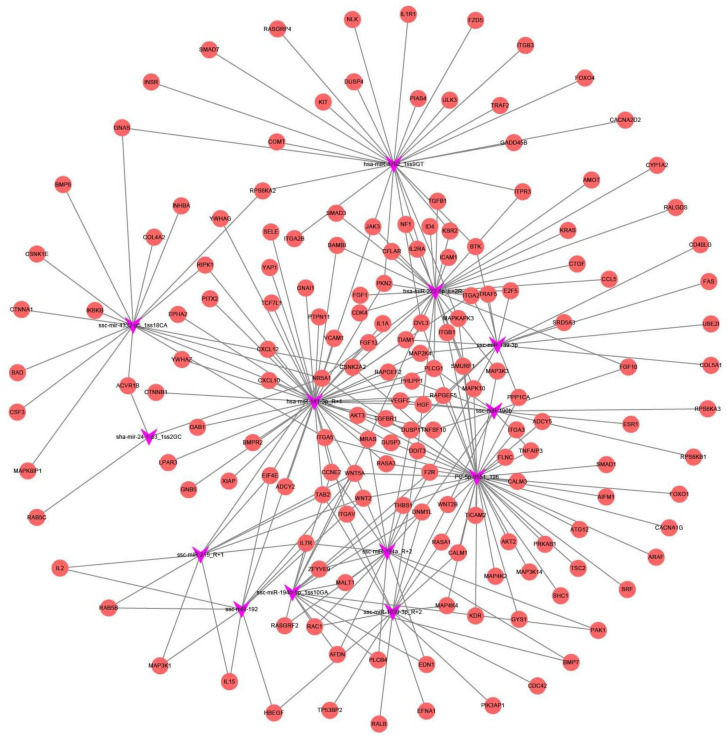
DEmiRNA–Gene network analysis. The miRNA–Gene network was built based on the interactions between miRNAs and the intersected target genes. This analysis illustrates the key regulatory functions of the identified miRNAs and their target genes. For the network, the diamond nodes represent miRNA, and the orange circles represent genes.

**Figure 6 cells-11-02362-f006:**
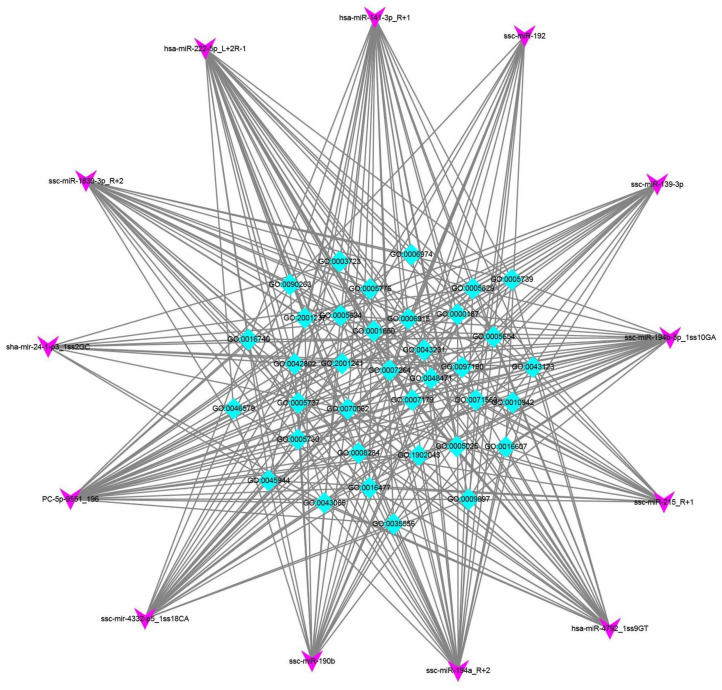
DEmiRNA–GO network analysis. The miRNA–GO network was built based on the relationship between significant biological functions and DEmiRNAs. The squares represent GOs affected by miRNAs, and the diamond nodes represent miRNA.

**Figure 7 cells-11-02362-f007:**
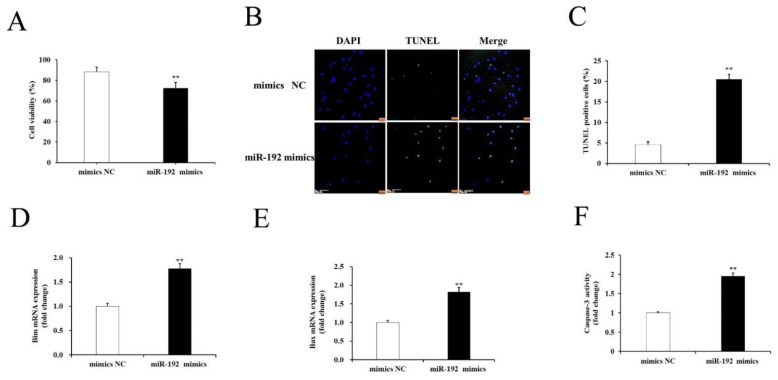
Overexpression of miR-192 induces PGCs apoptosis. (**A**) Effects of miR-192 overexpression on the proliferation of PGCs. (**B**) Apoptosis of GCs was evaluated using TUNEL assay; scale bars correspond to 50 μm. (**C**) Quantification of TUNEL-positive cells in porcine GCs. (**D**) The expression analysis of Bim. The relative expression data were normalized to β-actin. (**E**) The expression analysis of Bax. (**F**) The analysis of caspase-3 activity. The *t*-test was used to examine the differences between the groups. Data are presented as the means ± SE. (*n* = 3). ** *p* < 0.01.

**Figure 8 cells-11-02362-f008:**
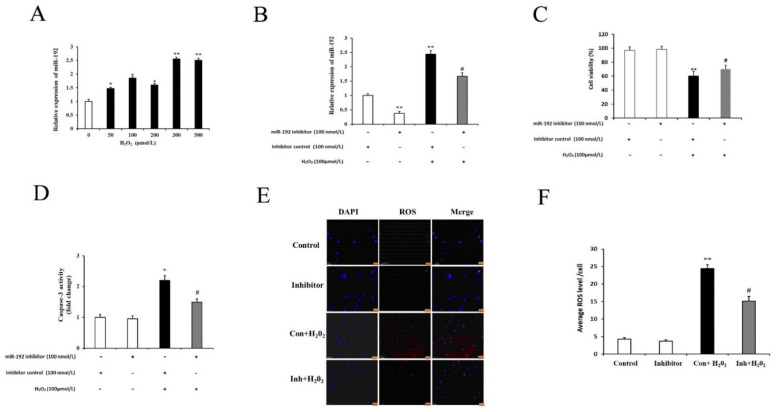
Downregulation of miR-192 alleviates H_2_O_2_-induced PGC injury. (**A**) The expression levels of miR192 in cultured PGCs were measured after 12 h of exposure to different H_2_O_2_ concentrations. (**B**) Transfection with miR-192 inhibitor decreased PGC miR-192 levels significantly. (**C**) Effects of miR-192 downregulation on PGC proliferation. (**D**) Effects of miR-192 downregulation on caspase-3 activity. (**E**) Effects of miR-192 downregulation on intracellular ROS levels, scale bars correspond to 50 μm. (**F**) Quantification of intracellular ROS levels. Each value is expressed as the mean ± S.E. (*n* = 3). * *p* <0.05 and ** *p* < 0.001 vs. con; # *p* < 0.01 Inh + H_2_O_2_ vs. con+ H_2_O_2_.

**Figure 9 cells-11-02362-f009:**
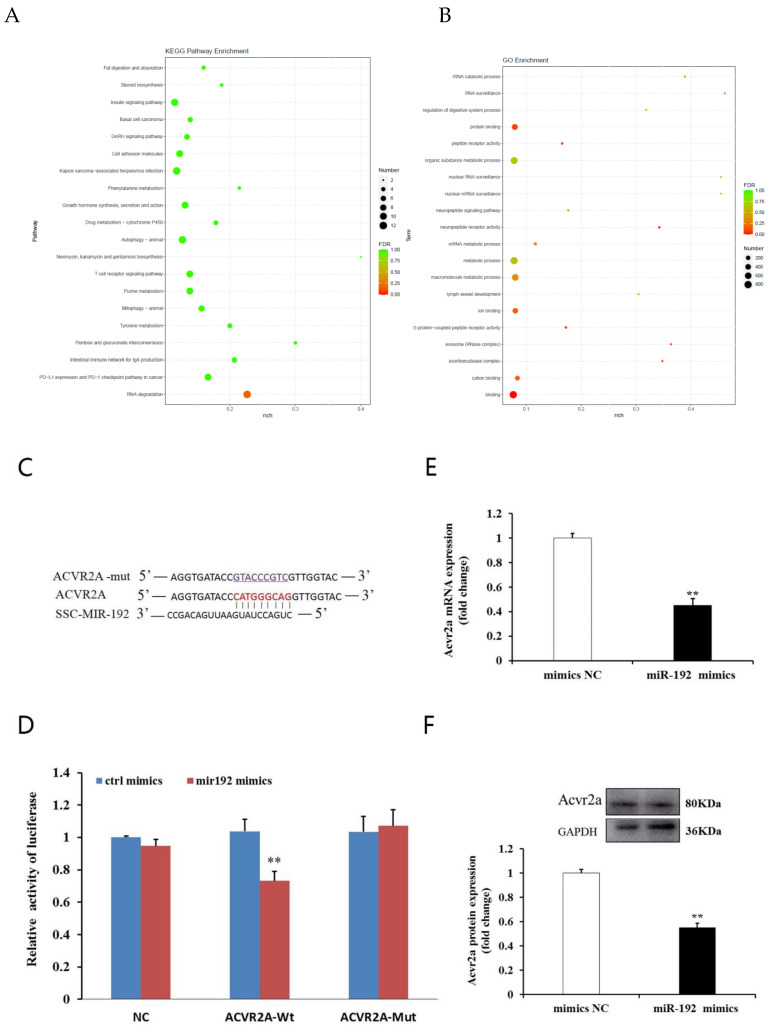
Identification of the miR-192 target gene. (**A**) Bubble chart indicating the KEGG pathways associated with miR-192 targets. (**B**) GO enrichment analysis of miR-192 targets. (**C**) The predicted binding region between miR-192 and Acvr2a mRNA was predicted using bioinformatics analysis. (**D**) Luciferase activity assay. (**E**) The mRNA expression of Acvr2a was assessed in GCs transfected with miR-192 mimics via RT-PCR. (**F**) At 48 h post-transfection, ACVR2A protein level in cultural PGCs was determined via Western blotting. GAPDH expression levels were used to normalize protein levels. Data are presented as means ± SE. (*n* = 3). ** *p* < 0.01.

**Figure 10 cells-11-02362-f010:**
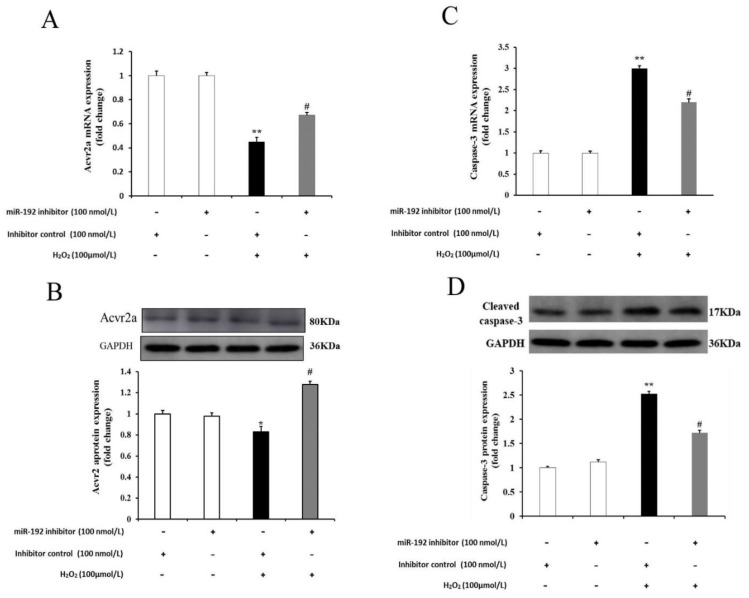
Downregulation of miR-192 attenuates H_2_O_2_-induced oxidative injury in PGCs through the regulation of Acvr2a and caspase-3. (**A**) Effects of miR-192 downregulation on the mRNA expression of Acvr2a. (**B**) Effects of miR-192 downregulation on the protein expression of ACVR2A. (**C**) Effects of miR-192 downregulation on the mRNA expression of caspase-3. (**D**) Effects of miR-192 downregulation on the protein levels of cleaved caspase-3. Data are shown as mean ± SE. (*n* = 3). * *p* < 0.05, ** *p* < 0.01 vs. the H_2_O_2_-free group (control); # *p* < 0.05 vs. the H_2_O_2_ plus inhibitor control.

**Figure 11 cells-11-02362-f011:**
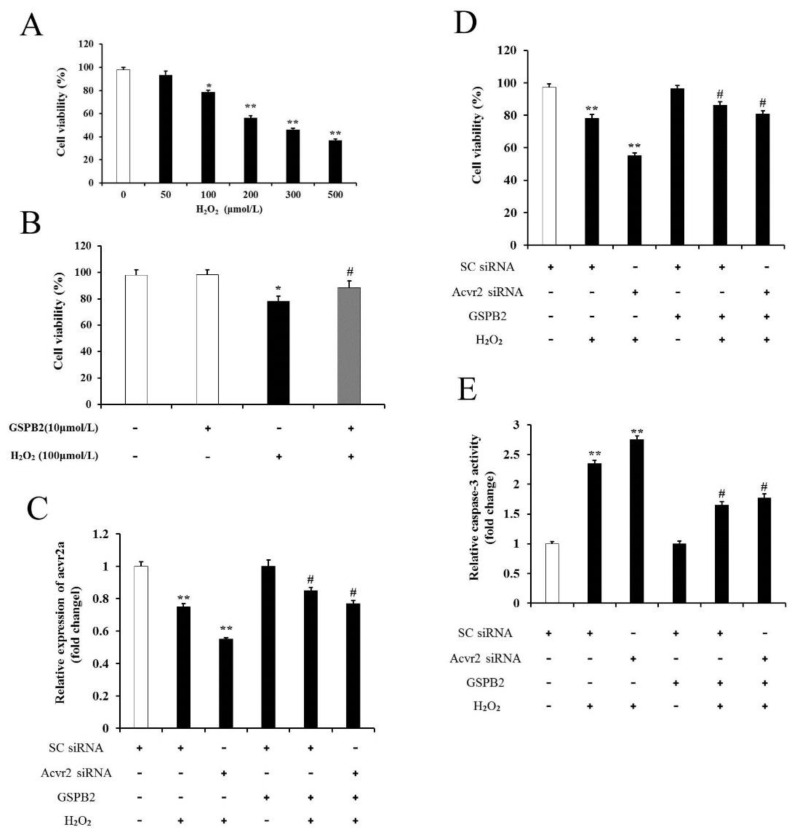
Acvr2a involved in GSPB2-mediated PGC protection during oxidative stress. (**A**) Primary cultured PGCs were treated with different concentrations of H_2_O_2_ for 12 h. Cell viability was then determined using CCK-8 assay. (**B**) PGCs were pretreated with or without GSPB2 (10 μM) for 12 h, rinsed in PBS, and then, incubated with 100 μM H_2_O_2_ for 12 h. Cells were then collected for measurement of cell viability. * *p* < 0.05, ** *p* < 0.01 vs. control group; # *p* < 0.05 vs. the H_2_O_2_-treated group. Data are shown as mean ± SE (*n* = 3). (**C**) PGCs transfected with Acvr2a siRNA or scrambled control siRNA for 24 h were cultured for another 12 h in the presence or absence of 10 μM GSPB2, and then, incubated with 100 μM H_2_O_2_ for 12 h. The mRNA expression of Acvr2a was determined via qRT-PCR. (**D**) Cell viability was determined using the CCK-8 assay. (**E**) Caspase-3 activity was measured using ELISA. Data are shown as mean ± SE. (*n* = 3). ** *p* < 0.05 vs. SC siRNA-alone group; # *p* < 0.05 vs. SC siRNA plus GSPB2 pretreatment.

**Figure 12 cells-11-02362-f012:**
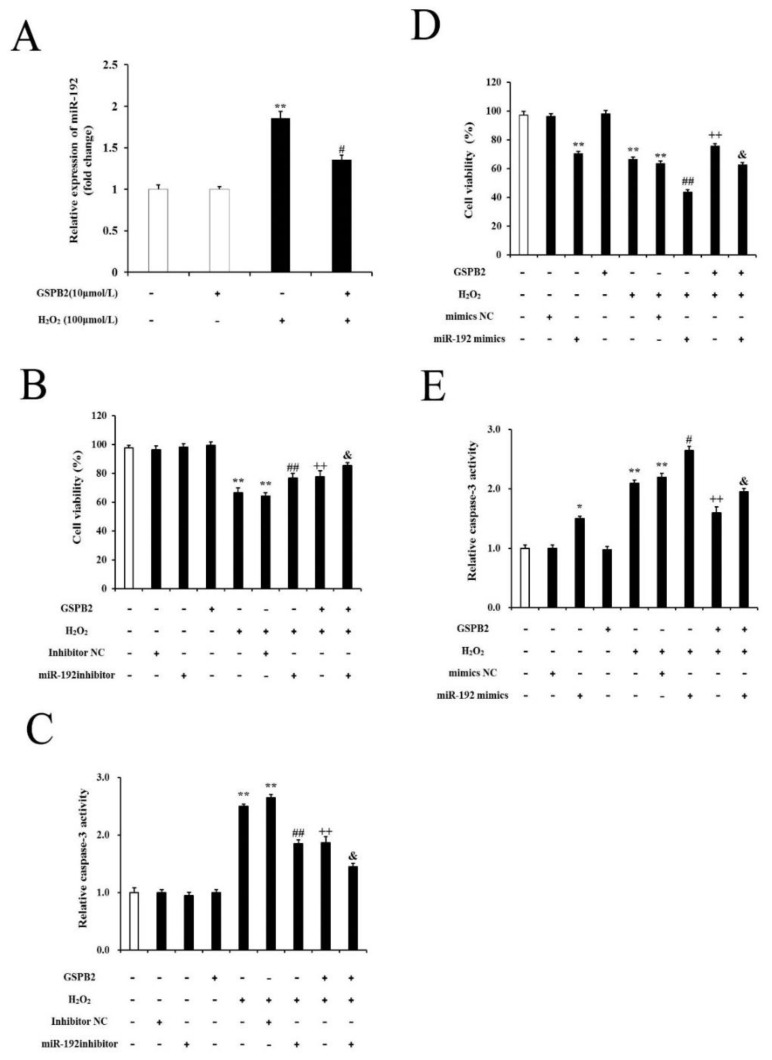
GSPB2 counteracts H_2_O_2_-induced PGC injury through the regulation of miR-192. (**A**) PGCs were pretreated with or without GSPB2 (10 μM) for 12 h, rinsed in PBS, and then, incubated with 100 μM H_2_O_2_ for 12 h. The expression of miR-192 was determined via qRT-PCR. Data are shown as mean ± SE. (*n* = 3). ** *p* < 0.01 vs. the control; ## *p* < 0.05 vs. H_2_O_2_ group. (**B**) PGCs were transfected with miR-192 inhibitor for 24 h; then, they were treated with or without GSPB2 (10 μM) for 12 h, and finally, incubated with H_2_O_2_ (100 μM) for 12 h. A CCK-8 assay was used to determine PGC viability. (**C**) PGCs were treated as above. The detection of caspase-3 activity was conducted via ELISA assay. (**D**) PGCs were transfected with miR-192 mimic for 24 h; then, they were treated with or without GSPB2 (10 μM) for 12 h, and finally, incubated with H_2_O_2_ (100 μM) for 12 h. PGC viability was determined via CCK-8 assay. (**E**) PGCs were treated as above. Caspase-3 activity was measured via ELISA assay. Data are shown as mean ± SE. (*n* = 3). ** *p* < 0.01 vs. the control; ## *p* < 0.01 vs. H_2_O_2_ plus miR-192 inhibitor; ++ *p* < 0.01 vs. GSPB2-alone group; & *p* < 0.05 vs. the GSPB2-plus-H_2_O_2_ group. * *p* < 0.05, # *p* < 0.05.

## Data Availability

The sequence data from this study have been submitted to the NCBI Gene Expression Omnibus (GEO) under the accession number GSE201369.

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
