# Peer review of "Downregulation of miR-192 Alleviates Oxidative Stress-Induced Porcine Granulosa Cell Injury by Directly Targeting Acvr2a"

_cells, 2022, doi:10.3390/cells11152362_

Round 1

Reviewer 1 Report

The work is very well designed. The study was carried out with granulosa cells (GCs) collected from healthy porcine ovarian follicles and treated with H2O2 for 12 h. In some experiments, cells were transfected with miRNAs (miR-192 mimics, miR-84 192 inhibitor, negative control (NC) mimics and inhibitor NC). Several analyzes were performed: cell viability, TUNEL assay, caspase-3 activity assay, mRNA and protein expression analyses, bioinformatic analysis. All figures are representative of the main results obtained. The bioinformatics analysis is very well explored by the authors highlighting the main findings with miRNAs. Through several methodological strategies, the authors show that reduction of miR-192 levels alleviates oxidative stress by directly targeting ACVR2A, a regulator of follicle development, in H2O2-induced porcine GC damage. This study also shows that GCs pretreated with GSPB2, a component of grape seed proanthocyanidin extract, markedly reduced caspase-3 activity induced by H2O2, which was associated with the increased and decreased expression of ACVR2A and miR-192, respectively.

Overall, the work is well written, includes a pertinent bibliography and presents a potential therapeutic approach to reproductive diseases treatment. However, below are some points that need to be better checked before final publication.

Minor points:

  • Material and Methods:

  1. p. 2, line 83: “…GCs were treated with 0 to 500…”. According to figure 1A, GCs were treated with 0 to 1000 μM. Please, check.

  2. Include methods for ROS analysis.

  3. Include western blot analysis for target proteins shown in the figures 9 and 10.

  4. Add information about mRNA expression analysis of ACVR2A and caspase-3.

  • Figure 3A: correct the word "number" in the description of the subtitle "nubmer of genes"

  • Check the ACVR2A nomenclature in the manuscript (in some parts it appears with lowercase letters, in others with capital letters).

Author Response

Response to Reviewer 1 Comments

Point 1: p.2.line 83: “…GCs were treated with 0 to 500…”. According to figure 1A, GCs were treated with 0 to 1000 μM. Please, check.

Response 1: The relevant content has been revised in the Material and Methods (See Line 83, Page 2).

Point 2: Include methods for ROS analysis.

Response 2: The relevant content has been added in the Material and Methods (See Line 94-102, Page 2-3).

Point 3: Include western blot analysis for target proteins shown in the figures 9 and 10.

Response 3: The relevant content has been added in the Material and Methods (See Line 185-197, Page 4).

Point 4: Add information about mRNA expression analysis of ACVR2A and caspase-3

Response 4: The relevant content has been added in the Material and Methods (See Line 165-171, Page 4; Line 109-114, Page 3).

Point 5: Figure 3A: correct the word "number" in the description of the subtitle "nubmer of genes"

Response 5: The relevant content has been revised in the Figure 3A (See Page 7)

Point 6: Check the ACVR2A nomenclature in the manuscript (in some parts it appears with lowercase letters, in others with capital letters).

Response 6: The ACVR2A nomenclature has been revised in the manuscript.

Reviewer 2 Report

The manuscript by Zhang et al. examines the impact of H2O2-mediated OS in PGCs thorough miRNA-seq. The authors identify different miRNA expressions depending on the concentration of H2O2 compared to controls. After some informatics analyses, they focus their research in one of those miRNAs, miR-192. miR-192 is the most induced by H2O2-mediated OS and its downregulation alleviates PGCs oxidative injury. They prove that miR-192 directly targeting on acvr2a and its downregulation decrease the Caspase 3 activity. Finally, they use a GSPB2, a compound that recent findings show that can alleviate oxidative stress-activated GC apoptosis.

The manuscript presents interesting results, it is well designed, written and the results support the conclusions. Under my point of view, the Ms. could be accepted in this Journal. Overall, it would be very interesting if the authors follow identifying more potential miRNAs therapeutic targets.

Additionally, I include a minor revision:

·       The authors sometimes use H2O2 and other times H2O2, they should use only one of those forms.

·       The author should describe in section 3.4 why they measure BIN and BAX expression.

·       Finally, it would be very interesting if the authors could make a hypothesis about the mechanism of miR-192 can modify the activity of the Caspase 3.

Author Response

Point 1: The authors sometimes use H2O2 and other times H2O2, they should use only one of those forms.

Response 1: The relevant content has been revised in the revised version.

Point 2: The author should describe in section 3.4 why they measure BIN and BAX expression

Response 2: The relevant content has been added in the results (See Line 459-462, Page10).

Point 3: Finally, it would be very interesting if the authors could make a hypothesis about the mechanism of miR-192 can modify the activity of the Caspase 3.

Response 3: The relevant content has been added in the revised version (See Line 952-956, Page 19-20).